# A Contemporary View of Respiratory Syncytial Virus (RSV) Biology and Strain-Specific Differences

**DOI:** 10.3390/pathogens8020067

**Published:** 2019-05-21

**Authors:** Mansi C. Pandya, Sean M. Callahan, Kyryll G. Savchenko, Christopher C. Stobart

**Affiliations:** Department of Biological Sciences, Butler University, Indianapolis, IN 46208, USA; mpandya@butler.edu (M.C.P.); smcallah@butler.edu (S.M.C.); ksavchen@butler.edu (K.G.S.)

**Keywords:** respiratory syncytial virus, RSV, strain-specific differences, viral evolution

## Abstract

Respiratory syncytial virus (RSV) is a human respiratory pathogen which remains a leading viral cause of hospitalizations and mortality among infants in their first year of life. Here, we review the biology of RSV, the primary laboratory isolates or strains which have been used to best characterize the virus since its discovery in 1956, and discuss the implications for genetic and functional variations between the established laboratory strains and the recently identified clinical isolates.

## 1. An Overview of RSV Biology

### 1.1. RSV Disease and Vaccines

Respiratory syncytial virus (RSV) is a human respiratory virus that is responsible for the majority of acute lower respiratory tract infections in infants worldwide [1]. In addition, it continues to be commonly associated with lower respiratory infections and mortality among the elderly and in individuals with compromised immune states [2,3,4,5,6]. In 2015, an estimated 33.1 million cases of acute lower respiratory infections that occurred worldwide were associated with RSV [7]. Of those cases identified, approximately 10% were associated with hospitalization and 0.4% resulted in mortality. Although RSV infections occur regularly in healthy infants with no predisposing factors, conditions associated with enhanced susceptibility for RSV disease include premature birth, lack of initial infection, lack of or limited levels of maternal antibodies for protection against RSV, the presence of certain genetic predisposing traits, and/or the presence of cardiopulmonary disease [8,9,10,11,12,13]. Despite a high global burden of disease being attributable to RSV, there remains no commercially-available vaccines for RSV. 

The World Health Organization (WHO) has identified RSV as a primary target for future vaccine development and has begun to establish standards for comparison. While many existing vaccines target a single primary population, the development of vaccines for RSV will likely need to target three distinct populations: Young infants (within the first three to six months of life), older infants and toddlers, and the elderly [2]. Numerous vaccine approaches are currently being explored to address one or more of these target populations, including live-attenuated, chimeric, vector-based, particle-based, subunit, and monoclonal antibody-based vaccine platforms [2]. The continued study of existing and developing strains of RSV will be essential to the future development of RSV vaccines, in order to understand the evolution of the virus and any potential changes in antigenicity that may arise. 

### 1.2. RSV Genome

RSV belongs to the order *Mononegavirales* and family *Pneumoviridae*. It is an enveloped virus with a non-segmented, negative-sense RNA genome (-ssRNA) that is approximately 15.2 kb in length [13]. The genomic organization of RSV consists of 10 genes which collectively encode 11 proteins (Figure 1). RSV forms filamentous enveloped virions that are primarily covered in the attachment (G) and fusion (F) glycoproteins, with a lesser amount of integrated pentameric small hydrophobic (SH) proteins within the envelope structure [13,14]. RSV F and G (see below) are responsible for mediating viral attachment and entry into cell hosts. Directly beneath the envelope, the viral matrix (M) protein plays a key role in the assembly and stability of virion structures [13,15]. Within the virion are the nucleoprotein (N), phosphoprotein (P), and RNA-dependent RNA polymerase (L), which protect the viral genome and mediate replication, along with a transcription processivity factor (M2-1) [13]. The two most common reverse genetics systems, which each employ the A2 strain genetic background, necessitate co-expression of these four viral proteins with a full-length antigenomic construct for infectious clone assembly [15,16]. In addition to the aforementioned structural genes and replicative machinery, RSV also produces two nonstructural proteins (NS1 and NS2), which have been shown to suppress innate immune signaling and antagonize apoptotic pathways [17,18,19,20].

### 1.3. RSV Antigenicity and Infectivity

RSV G and F are arguably the most well-studied proteins of the virus due to their critical roles in mediating attachment and fusion, respectively, as well as being responsible for inducing the majority of neutralizing antibodies in vivo [21,22,23,24]. RSV G is produced in membrane-bound and secreted forms during infection [25,26,27,28]. The membrane-bound form of RSV G is heavily glycosylated and exhibits two highly variable mucin-like regions and a central conserved region [29,30,31,32]. Overall, RSV G exhibits the most variable RSV gene sequence and has been used in the characterization of virus evolution and to establish genetic variants of circulating RSV [33]. While several host proteins have been identified as possible targets of RSV attachment in a wide range of cells, including heparin sulfate, surfactant protein A (SP-A), the fractalkine receptor (CX3CR1), and annexin II, it remains unclear what the primary host target for RSV G-mediated attachment is during infection in the human airway epithelium [34,35,36,37,38,39]. This initial interaction between RSV G and the host cell facilitates and likely aids in the engagement of RSV F to drive fusion between the viral envelope and host membrane [40]. During fusion, RSV F undergoes a dynamic conformational change from a metastable pre-fusion trimer to a stable post-fusion state [14,33,41,42]. While several host receptors for F attachment have been proposed, including nucleolin and the epidermal growth factor receptor (EGFR), the possible portals of RSV entry and the potential receptors for host fusion remain unclear [43,44,45]. Studies evaluating neutralizing antibodies of RSV F have identified several neutralizing sites, with the most potent neutralizing antibodies being associated with the binding of the prefusion conformational state of RSV F [24,41,46]. Subsequently, most current vaccine design efforts are focused on optimizing RSV F as a platform for inducing immunologic protection [2,47].

### 1.4. RSV Genetic Diversity

Traditionally, RSV is classified into two distinct groups or subtypes, RSV-A and RSV-B, which diverged approximately 350 years ago and are based on antigenic and sequence-based variations predominately associated with RSV G [48,49,50,51]. RSV exhibits seasonality with multiple genotypes, often in co-circulation with a dominance shift between RSV-A and RSV-B types every one to two years [52]. Within each of these two groups, several genotypes have been identified and described [49,50,53,54]. Recently, several unique genetic modifications in RSV G have been identified, which include a 72-nucleotide duplication (referred to as the ON genotype) associated with RSV-A types and a 60-nucleotide duplication (referred to as the BA genotype) associated with RSV-B types [53,55,56,57]. In the cases of each of these two new genotypes, they have rapidly become the predominant forms found in circulation worldwide and appear to increase in vitro viral fitness and attachment [58]. Furthermore, variations in RSV-A and RSV-B over time appear to correlate with the induction of anti-G monoclonal antibodies that recognize the primary epitopes, thus indicating immune-driven RSV evolution [59].

## 2. Common Laboratory Strains of RSV

Since its initial isolation from a chimpanzee in 1956 during an outbreak of coryza in a colony of animals and subsequent discovery in humans shortly after, most research on RSV has centered on the use of a limited number of historical isolates [60,61,62]. These prototypic “laboratory strains”, while playing critical roles for identifying the key functions of viral proteins and the general virology of RSV infection, have shown in limited studies to exhibit subtle to significant differences in cytopathology, antigenicity, and pathogenicity, both in vitro and in vivo, when compared to other laboratory strains and clinical isolates. In this section, we will review the origins, general biology, and use of several common prototypic laboratory strains.

### 2.1. RSV Long

The RSV Long strain was first isolated by Robert Chanock from a child with bronchopneumonia in 1956 and was subsequently passaged 11 to 13 times in HEp-2 cells [62]. As the “first” prototypic strain, the Long strain was used extensively in the 1960s and 1970s to characterize the initial physical properties of the virus, antigenic variations between RSV isolates, pathophysiology, and epidemiology of RSV disease [60,63,64,65,66,67,68,69,70]. The RSV Long strain has also played an important role in the establishment of early animal models of RSV infection in vivo, including mice, ferrets, and cotton rats [71,72,73]. Although existing platforms using the genetic background of A2 had been available, in 2014, the first reverse genetics platform based on the Long genetic background was developed [74].

An early study in 1963 that compared the antigenic responses to strains Long and CH-18537 (described below) was pivotal in suggesting that antigenically distinct strains of RSV exist [66]. Later studies would classify these strains into subtypes (groups) A and B, respectively, due in large part to the antigenic differences in the attachment glycoprotein G [48,75]. Along with strain A2 (described below), the Long strain continues to be recognized as a prototypic group (or subtype) A virus and is regularly used in comparative studies of antigenicity and neutralization [76,77]. 

### 2.2. RSV A2

RSV strain A2 was first isolated in 1961 from the lower respiratory tract of an infant in Melbourne, Australia [78]. Since its initial isolation, RSV A2 has been established as the prototypic A strain for the study of RSV and remains the most common platform for the development of RSV live-attenuated vaccine candidates.

Although not recapitulating human disease, numerous studies have employed the A2 strain to elucidate and better characterize the immune responses to RSV in both mice and cotton rats [63,79,80,81]. In several common cell culture lines for the propagation and study of RSV (HEp-2, Vero, A549 cells) and in airway epithelial cultures, RSV A2 is generally associated with higher replication kinetics and more cytopathic effects earlier in infection when compared to other “laboratory” or clinical isolates [42,81,82,83]. It has been hypothesized that this may be due to a long history of passage-based adaptation in these cell lines [82]. The flexibility and ease by which A2 is able to replicate efficiently in a variety of cell platforms has also led many to believe that the virus utilizes a wide array of different receptors and attachment factors, including heparin sulfate and nucleolin [38,82]. While no significant morphological differences in virion structure or organization were observed between A2 and a clinical isolate (A/TN/12/11-19), studies have reported that RSV A2 exhibits less thermal stability and potentially less pre-fusion F availability when compared to strain A2-line19F, further implicating that appreciable differences in stability and antigenicity may exist at a molecular level, even between similar strains [14,42,84]. 

A key factor in the use of A2 in RSV research has been the establishment of two separate reverse genetics platforms based on the A2 genetic background. In 1995, the first reverse genetics system for RSV was developed through the co-transfection of a plasmid encoding the antigenome, along with supplementary plasmids expressing N, P, M2-1, and L proteins [16]. While this initial system was not very efficient, it did yield a platform to recover mutant infectious viruses and opened the door to the development of some of the first recombinant live-attenuated vaccines based on the A2 genetic background. In 2012, a similar reverse genetics platform was developed which utilized a bacterial artificial chromosome (BAC) for expression of the antigenome and was associated with more efficient virus recovery [15]. While the 1995 system was initially based on the A2 genetic background, the 2012 system was initially engineered to generate an A2-line19F virus. Studies comparing A2 and A2-line19F showed that the F protein of strain Line19 (see discussion below) was associated with more pathogenesis in mice when compared to strain A2 [83]. 

In summary, A2 continues to be a prototypic RSV strain and has been used extensively as a reverse genetics platform for the development of the majority of live-attenuated vaccine candidates to date. The strain is a key system for the study of RSV structure and has played a key role in elucidating immune responses in animal models to RSV infection [79].

### 2.3. RSV Line 19 and A2-line19F

Line 19 is an RSV subtype A virus that was first isolated from an infant with respiratory illness at the University of Michigan Hospital in 1967 and initially expanded in WI-38 cells [85]. However, subsequent studies of genomic differences between the Line 19 and Long strains have suggested that Line 19 may be a Long strain, mouse-adapted through serial intracranial inoculations in suckling mice generated in the same laboratory [73,79]. The pathogenesis of Line 19 infection in mice is distinct and different from other common strains such as A2 and Long. Infections in mice with strain Line 19 are biased towards a more T_H_2-type antiviral response, with increased goblet cell expansion and elevated IL-13 and MUC5AC levels when compared to the T_H_1 responses associated with A2 and Long [86,87]. Compared to strain A2, Line 19 is also associated with lower overall titers in both HEp-2 cells and in mice in vivo [86]. A follow-up study using a recombinant A2-line19F virus (A2 expressing the F protein of Line 19) showed that the fusion protein of strain Line 19 was genetically unique and responsible for the differential immunopathology observed between the strains [83]. Regardless of its origin, strain Line 19 and a related strain, A2-line19F, have been established as important mouse models of pulmonary pathophysiology and immunologic responses.

### 2.4. RSV CH-18537

CH-18537 is a RSV subtype B virus that was first isolated from a throat swab of a child with upper respiratory disease in 1962. The virus initially expanded in Wistar 26 cells [66]. Early studies in the 1960s established that CH-18537 differed in antigenicity from the other contemporary isolates at the time [66]. Since these initial studies, CH-18537 has been established as a common prototype RSV-B strain for genetic studies and continues to be used for evaluating antigenicity and genetic diversity [75,88,89,90,91,92].

### 2.5. RSV Memphis-37

The Memphis-37 strain is an RSV subtype A virus isolated as a nasal aspirate in 2001 from a 4-month-old African American male child with bronchiolitis [93]. The initial isolate was plaque purified and expanded as a GMP-lot in FDA-approved Vero cells for use in human clinical trials. Memphis-37 has been used to study RSV pathogenesis, the dynamics of immunologic responses to infection in healthy immunocompetent adults, and as a platform for testing inhibitors, vaccines, and therapeutics for RSV infection [94,95,96,97]. Intranasal inoculation of neonatal lambs with Memphis-37 has been shown to better reflect the pathogenesis and immune responses seen in humans when compared to other animal models [98]. Memphis-37 remains a primary model to study RSV pathogenesis and immunologic responses in humans.

## 3. Contemporary RSV Strains, Pathogenesis, and Viral Evolution

### 3.1. The Genetics and Evolution of RSV Clinical Isolates

Contemporary RSV clinical isolates are initially classified into either RSV-A or RSV-B subgroups, then often further classified into one of a rapidly growing number of genotypes [49,50,53,56,99]. Different genotypes are known to co-circulate during RSV seasons and genotypic dominance can vary based on the year and location [99,100]. Epidemiological studies often reveal genetic (and sometimes phenotypic) variance between isolates of the same subgroup and genotype [101]. The continual emergence of new strains and the identification of new genotypes highlights the ongoing evolution of the virus worldwide.

In the last 20 years, there have been two new emergent genotypes which have each taken precedence: ON1 and BA. The BA genotype emerged in samples obtained in Buenos Aires in 1999. These viruses exhibited a 60-nucleotide duplication within a hypervariable region in the G gene of RSV-B and within a few years became the predominant form of RSV-B detected worldwide [50,102,103,104,105]. Analysis of BA strains in vitro has shown that the G duplication impacts both viral attachment and replication, providing a fitness advantage over viruses lacking the duplication [58]. In parallel to the emergence of the G duplication in RSV-B strains, the more recent emergence and predominance of the RSV-A ON1 genotype over NA1 (its precursor lineage lacking the G duplication) has been well-noted and described [55,106,107]. During the RSV season of 2010–2011 in Ontario, Canada, a novel RSV genotype (ON1) containing a 72-nucleotide duplication in the C-terminal region of RSV G was identified, which resulted in up to seven potential additional O-glycosylation sites within the protein. Similar to the BA genotype, the ON1 genotype quickly become common worldwide among RSV-A isolates [56,108,109]. Several recent groups have started describing mutations in RSV G which result in premature stop codons and deletions of the ON1 and BA genotypes [110]. These studies may indicate a growing level of immunity within some populations.

Analysis of the RSV F and G of recent isolates from 2015–2017 in the United States has shown not only that changes in RSV G were continuing to occur, but also changes in known antigenic sites in RSV F [109]. Recent analysis of isolates in Kenya has shown that adaptive amino acid changes are also being detected in other viral genes, including the viral polymerase (L) and M2-1 proteins [111]. A recent study from Lebanon has identified two additional RSV-A genotype variants (LBA1 and LBA2), but has also indicated a selection of variants with increased resistance to palivizumab, the only available prophylaxis option worldwide for RSV [112]. While nearly all studies evaluating recent clinical isolates have focused on genetic variations (rather than phenotypic differences), these viruses are known to differ appreciably in genetic sequence from current laboratory strains. Several studies have been performed to determine the relative mutation rates of RSV strains and have shown rates ranging from 10^-3^ to 10^-4^ nucleotide substitutions/site/year, depending upon the location and strain [49,50,113,114]. These data strongly suggest that evolutionary pressures are continuing to drive RSV evolution and potentially push modern circulating RSV strains further from the common laboratory strains isolated during 1950s and 1960s. We performed a phylogenetic analysis of RSV G gene sequences, comparing common laboratory strains and contemporary clinical isolates (Figure 2). From our analysis, and those performed by others, laboratory strains tend to cluster out and separate from contemporary clinical isolates, indicating that the viruses have likely evolved not only in genetic space, but potentially also in phenotype [49,50,53,57]. 

### 3.2. Disease, Pathogenesis, and Cytopathology among Clinical Isolates

Numerous recent studies have surveyed clinical isolates obtained from sites around the world. From these multiyear studies, both the RSV-A and RSV-B strains co-circulate, with the dominance of strains varying over time [115,116,117]. However, RSV-A infections appear more frequent and appear to be associated with higher transmissibility than RSV-B [118,119,120]. The persistence of both of these distinct groups may explain the ability of previously infected individuals to remain susceptible to RSV infections [121]. While several studies have evaluated the disease potential of obtained RSV clinical isolates, the relationships and extent of variation, with regards to genetic sequences, genotypes, and clinical severities, remain unclear [122]. Confounding these issues, several multiyear studies have indicated significant phenotypic differences among the surveyed clinical isolates, while other studies have reported that no significant differences were observed [115,116,117,122,123,124,125]. Furthermore, very few studies have begun to evaluate the replication and pathophysiology of clinical isolates. The ambiguity in available clinical data, along with a lack of replication and pathophysiology data highlights an extensive need for understanding the current biology of circulating strains of RSV.

The limited studies that have begun to evaluate clinical strain differences, both in vitro and in vivo, have highlighted significant differences among strains, even within the same genotype grouping. In vitro models of infection have historically relied upon infections in continuous cell lines including HEp-2, A549, BEAS-2B, and Vero cells, however recent studies have demonstrated that these cell lines fail to recapitulate the native architecture of the airway epithelium and may bias the entry, spread, and infectivity of RSV [126,127,128,129]. A study using primary pediatric bronchial epithelial cells (PBECs) investigated whether the replication and cytopathology of prototypic laboratory strain A2 was representative of a panel of four recent clinical isolates. In this study, Villenave et al. demonstrated that A2 exhibits markedly different cytopathology, viral titers, and cytokine secretion than the clinical isolates surveyed [82]. Hypersecretion of mucus is a hallmark of severe neonatal RSV infections and is associated with higher disease potential in the lower respiratory tract [130]. In another study by Stokes et al., an analysis of a panel of six clinical isolates demonstrated significant differences among strains in disease severity, lung IL-13 cytokine production, and mucus production during infections in mice [81]. As previously described, studies have shown that the G duplications associated with the ON1 and BA genotypes may confer increased fitness, both in vitro and in vivo, over those lacking these modifications, yet none of the prototypic laboratory strains express these important genetic variations. Collectively, these studies appear to suggest that conventional laboratory strains of RSV may not best represent the biology and pathology of clinical isolates circulating today.

## 4. Summary and Future Implications for Vaccine Design

As reviewed here, advances in our understanding of the biology of RSV continue to be driven through the extensive study of a limited number of prototypic laboratory strains. These studies have made great progress towards establishing the basis for current vaccine design efforts. However, while distinct differences have been observed (Table 1), far less is known on whether these “laboratory” models of RSV and contemporary clinical isolates differ significantly in infectivity, replication, or cytopathology. Extensive efforts are currently underway, by a multitude of groups, to develop effective vaccines for RSV. It remains unclear how well these candidate platforms will perform in providing broadly neutralizing coverage to circulating clinical isolates today and the near future. More effort should be afforded to the study of more recent isolates and to identify how these viruses fundamentally differ from the laboratory strains that have been characterized in the past. Understanding the extent of variations in genetics and phenotypes among RSV strains will inform better design of future vaccine constructs while also providing a more detailed picture of the landscape of RSV disease and pathogenesis in human populations today. 

## Figures and Tables

**Figure 1 pathogens-08-00067-f001:**
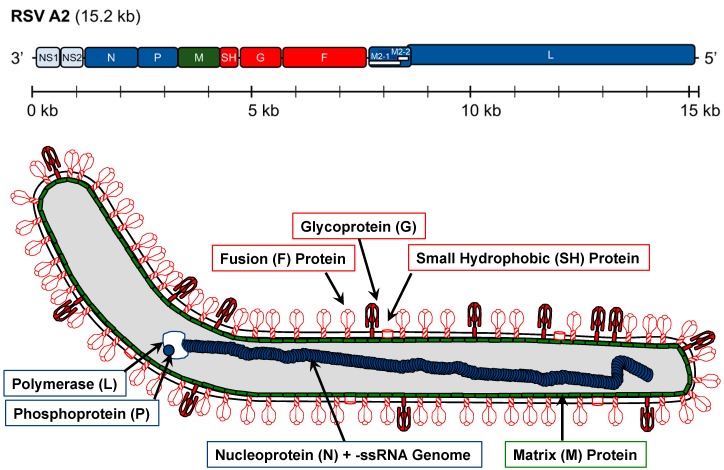
The genomic organization and virion structure of respiratory syncytial virus (RSV) strain A2 are shown with each of the 10 genes color-coded based on their relative functions: Immunomodulatory (light blue), envelope structure (green), surface structures for attachment and entry (red), and replication and genomic stability (dark blue). A size scale is provided with 1 kb gradations for evaluation of gene size. NS1/NS2, nonstructural protein 1/2; N, nucleocapsid; P, phosphoprotein; M, matrix; SH, small hydrophobic glycoprotein; G, attachment glycoprotein; F, fusion glycoprotein; L, large polymerase protein (RdRP).

**Figure 2 pathogens-08-00067-f002:**
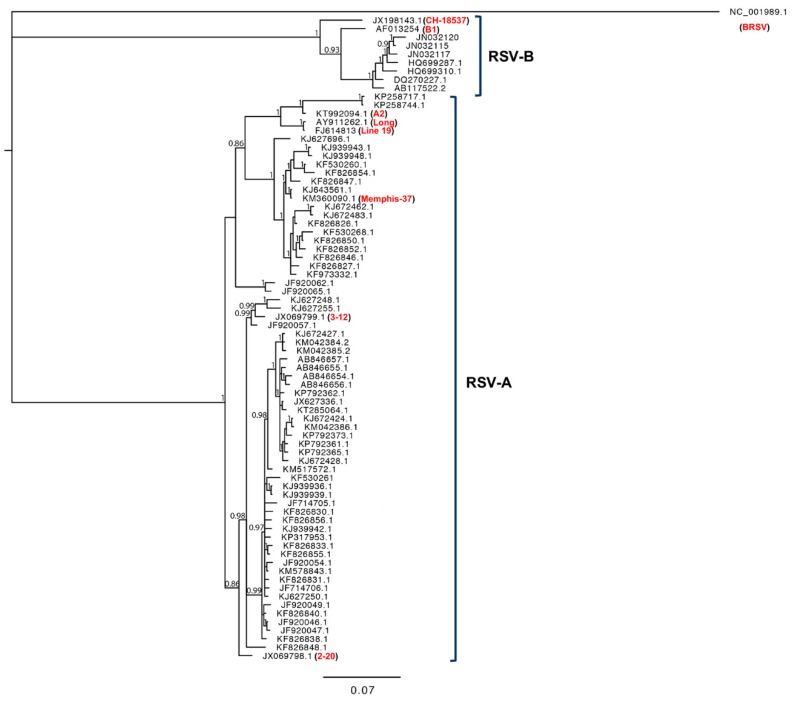
Phylogenetic analysis of an alignment of the RSV G gene sequences of common laboratory strains (identified in red) and contemporary clinical RSV-A and RSV-B isolates. A Bayesian inference of phylogenetic relationships between RSV G nucleotide sequences is shown. The phylogenetic outgroup was bovine RSV (BRSV). Numbers on branches are estimates for PPs (posterior probabilities) from the Bayesian inference (only numbers higher than 0.8 are shown). Common laboratory RSV-A strains A2, Long, Line 19, and Memphis (which are described in Section 2), as well as strains A2001/2-20, and A2001/3-12 (which have been more recently isolated and used to study RSV pathogenesis) are shown for phylogenetic comparison. Common laboratory RSV-B strains CH-18537 and B1 are also shown for comparison.

**Table 1 pathogens-08-00067-t001:** Comparative summary of laboratory and clinical isolates reviewed. The genetic type and common characteristics and applications are provided for each strain (or group).

Virus Strain	Type and Designation	Characteristics and Applications
Long	RSV-A (Laboratory Strain)	Isolated in 1956 Primarily used today in studies of antigenicity Prototypic RSV-A model
A2	RSV-A (Laboratory Strain)	Isolated in 1961 Most well-studied strain in use today; a prototypic RSV-A model Higher replication kinetics in vitro compared to other strains Mild cytopathology in animal models compared to other strains Most commonly employed reverse genetics system Most common strain used in live-attenuated vaccine preparations
Line 19/A2-line19F	RSV-A (Laboratory Strain)	Isolated in 1967 (A2-line19F synthesized in 2009) Primarily used in pathogenesis and immunology studies Lower viral load, but more severe pathophysiology in animal models Exhibits enhanced thermal stability compared to other strains
CH 18537	RSV-B (Laboratory Strain)	Isolated in 1962 Primarily used in antigenicity studies Prototypic RSV-B model
Memphis-37	RSV-A (Laboratory Strain)	Isolated in 2001 Produced as a GMP lot for clinical studies Primarily used in human pathogenesis and challenge studies
Clinical Isolates	RSV-A and RSV-B Isolates	Vary significantly in genetic diversity and differ from conventional laboratory strains Most studies to date have focused on genetic variations and epidemiology Many circulating types today exhibit a G protein duplication (ON1 and BA genotypes) Limited pathogenesis studies suggest variations in cytopathology and pathogenesis between isolates

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
