# Peer review of "A Contemporary View of Respiratory Syncytial Virus (RSV) Biology and Strain-Specific Differences"

_pathogens, 2019, doi:10.3390/pathogens8020067_

Round 1
Reviewer 1 Report
The review entitled "A Contemporary View of Respiratory Syncytial Virus 3 (RSV) Biology and Strain-specific Differences" gives an updated overview of RSV biology emphasizing on the primary laboratory isolates and the genetic and functional variations between established laboratory strains and recently identified clinical isolates.
1. Authors must update references using endnote (Ref 12)
Author Response
Reviewer 1
The review entitled "A Contemporary View of Respiratory Syncytial Virus 3 (RSV) Biology and Strain-specific Differences" gives an updated overview of RSV biology emphasizing on the primary laboratory isolates and the genetic and functional variations between established laboratory strains and recently identified clinical isolates.
1. Authors must update references using endnote (Ref 12)
Comments: We have updated this reference for consistency in format with the other references provided and thank the reviewer for their helpful recommendation.
Reviewer 2 Report
In the present manuscript, Pandya and collaborators review the differences between “classic” laboratory strains commonly used in RSV research and those that have circulated in the last two decades, and the potential implications of such divergence. The text is very well written in terms of structure, grammar and use of the English language, and focuses on a particularly interesting niche of RSV biology that underscores more detailed study. However, I consider there is a major point that needs to be addressed before considering the manuscript for publication, which will significantly improve the comprehension of the message to be conveyed for the expert and non-expert readership of Pathogens.
Major point:
1. Please include a comparative table summarizing the main characteristics of the different “common” and “contemporary” RSV strains described in sections 2 and 3, including a column with specific comments/notes (i.e. “prototypic A2 strain”, “particular Th2 response in mice”, etc). I believe this table will better showcase the complexity and the potential of the topic as well as foster critical thinking on the implications of the choice of different strains. Also, Figure 1 could be improved by adding a schematic representation of the viral particle, with different viral proteins following the color coding used for the genomic organization.
Minor points:
1. Although not mandatory, Figure 1 could also be completed by some specific visual support on the G and F proteins in order to better illustrate section 1.3. This could include one or many of the following: representation of protein structure, putative receptors, attachment/fusion functions, membrane-bound/secreted form, pre-fusion/post-fusion configuration, sequence variability, etc.
2. Is it possible to better identify the different RSV-A subtypes in Figure 2? Also, the rationale for the inclusion of 2001 clinical isolates 2-20 and 3-12 should be explained in the text and/or in the figure legend.
3. Please give some short background information to introduce the NA1 genotype mentioned in Line 210.
4. Please name the “two additional RSV-A genotypes” mentioned in Line 222.
Typo/text editing:
Line 30: please add “the” between “begun” and “establishment”.
Line 102: please add the missing " after “Laboratory Strains”.
Line 142: please add a “,” after “A2-line19F”.
Author Response
Reviewer 2
In the present manuscript, Pandya and collaborators review the differences between “classic” laboratory strains commonly used in RSV research and those that have circulated in the last two decades, and the potential implications of such divergence. The text is very well written in terms of structure, grammar and use of the English language, and focuses on a particularly interesting niche of RSV biology that underscores more detailed study. However, I consider there is a major point that needs to be addressed before considering the manuscript for publication, which will significantly improve the comprehension of the message to be conveyed for the expert and non-expert readership of Pathogens.
Major point:
1. Please include a comparative table summarizing the main characteristics of the different “common” and “contemporary” RSV strains described in sections 2 and 3, including a column with specific comments/notes (i.e. “prototypic A2 strain”, “particular Th2 response in mice”, etc). I believe this table will better showcase the complexity and the potential of the topic as well as foster critical thinking on the implications of the choice of different strains.
Comments: We appreciate the reviewer’s suggestion and agree completely. We have provided a table (see Table 1) which summarizes the characteristics and common applications of the laboratory strains with a comparison to the clinical isolates being obtained today.
2. Also, Figure 1 could be improved by adding a schematic representation of the viral particle, with different viral proteins following the color coding used for the genomic organization.
Comments: A schematic depicting the virion structure and matching the color-coding from the genome has now been provided (see Figure 1).
Minor points:
1. Although not mandatory, Figure 1 could also be completed by some specific visual support on the G and F proteins in order to better illustrate section 1.3. This could include one or many of the following: representation of protein structure, putative receptors, attachment/fusion functions, membrane-bound/secreted form, pre-fusion/post-fusion configuration, sequence variability, etc.
Comments: Based on the suggestion for a virion representing the locations and orientations of the genes above, we incorporated accurate representations of the pre-fusion F trimers and G glycoproteins (as best we understand them) in the virion depiction (see Figure 1).
2. Is it possible to better identify the different RSV-A subtypes in Figure 2? Also, the rationale for the inclusion of 2001 clinical isolates 2-20 and 3-12 should be explained in the text and/or in the figure legend.
Comments: A more clear description of the laboratory strains included in the study is now provided in the Figure 2 legend. In addition, a statement providing the rationale for inclusion of strains 2-20 and 3-12 is also now included in the figure legend provided.
3. Please give some short background information to introduce the NA1 genotype mentioned in Line 210.
Comments: A brief statement has been added to clarify the importance and significance of the NA1 lineage and its relatedness to the ON1 lineage (see page 5; lines 214 – 215)
4. Please name the “two additional RSV-A genotypes” mentioned in Line 222.
Comments: These genotypic variants are now described. See page 6; lines 239 – 240.
Round 2
Reviewer 2 Report
The authors have satisfactorily addressed all my concerns and the effort to improve the clarity of the message (modifications to Figure 1 and new Table 1) are greatly appreciated. I have only detected a few minor typo errors left to be corrected. Once these points are ok, the manuscript will be suitable for publication without further revision.
Typo/text editing:
Line 103: please replace “Laboratory with Laboratory (remove the “).
Line 290: please replace known of with known on.
Table 1 (Line 19 / A2 – line19F): remove Line 19 in the first bullet-point of the Characteristics and Applications in order to start directly with Isolated in 1967, as done for every other strain.
Author Response
We have made the minor revisions recommended by the reviewer and appreciate their assistance with finding these errors.